A methodological approach for inferring causal relationships from opinions and news-derived events with an application to climate change

Marten Juan 1
Delbianco Fernando 2 3
http://orcid.org/0000-0003-2988-4519 Tohme Fernando 2 3
http://orcid.org/0000-0003-4912-7961 Maguitman Ana G. 1 4 agm@cs.uns.edu.ar
1 Departamento de Ciencias e Ingeniería de la Computación, Universidad Nacional del Sur , Bahía Blanca, Buenos Aires , Argentina
2 Departamento de Economía, Universidad Nacional del Sur , Bahía Blanca, Buenos Aires , Argentina
3 Instituto de Matemática de Bahía Blanca , Bahía Blanca, Buenos Aires , Argentina
4 Instituto de Ciencias e Ingeniería de la Computación , Bahía Blanca, Buenos Aires , Argentina
Ahn Yong-Yeol
Electronic publication date: 2025 Jun 19
Publication date: 2025
Volume: 11
Electronic Location ID: e2964
Received 2024 Dec 12; Accepted 2025 May 27
Copyright: © 2025 Marten et al.
Copyright year: 2025
Copyright holder: Marten et al.
License: This is an open access article distributed under the terms of the Creative Commons Attribution License, which permits unrestricted use, distribution, reproduction and adaptation in any medium and for any purpose provided that it is properly attributed. For attribution, the original author(s), title, publication source (PeerJ Computer Science) and either DOI or URL of the article must be cited.
License URL: https://creativecommons.org/licenses/by/4.0/

Keywords: Causal analysis, Climate change, Opinion mining, Topic mining, Social media mining, Sentiment analysis, Stochastic causality

Funding: Universidad Nacional del Sur PGI-UNS 24/N060 and 24/E172 ANPCyT PICT 2019-01640, PICT 2019-02302, and PICT 2019-03944 Google Academic Research Award GARA 2024 This work was funded by Universidad Nacional del Sur (PGI-UNS 24/N060 and 24/E172), ANPCyT (PICT 2019-01640, PICT 2019-02302, and PICT 2019-03944), and a Google Academic Research Award (GARA 2024). The funders had no role in study design, data collection and analysis, decision to publish, or preparation of the manuscript.

==============================
Social media platforms like Twitter (now X) provide a global forum for discussing ideas. In this work, we propose a novel methodology for detecting causal relationships in online discourse. Our approach integrates multiple causal inference techniques to analyze how public sentiment and discourse evolve in response to key events and influential figures, using five causal detection methods: Direct-LiNGAM, PC, PCMCI, VAR, and stochastic causality. The datasets contain variables, such as different topics, sentiments, and real-world events, among which we seek to detect causal relationships at different frequencies. The proposed methodology is applied to climate change opinions and data, offering insights into the causal relationships among public sentiment, specific topics, and natural disasters. This approach provides a framework for analyzing various causal questions. In the specific case of climate change, we can hypothesize that a surge in discussions on a specific topic consistently precedes a change in overall sentiment, level of aggressiveness, or the proportion of users expressing certain stances. We can also conjecture that real-world events, like natural disasters and the rise to power of politicians leaning towards climate change denial, may have a noticeable impact on the discussion on social media. We illustrate how the proposed methodology can be applied to examine these questions by combining datasets on tweets and climate disasters.

Introduction

Social media has evolved into an invaluable source of information, capturing the opinions of millions of individuals globally. Among social media platforms, X, formerly known as Twitter, stands out as a unique channel where individuals can share their thoughts, emotions, and perspectives on a diverse range of subjects, from politics and public health to environmental issues. In addition to online discussions, alternative data sources such as historical records, economic indicators, and real-world events could also be examined collectively. To analyze these diverse data types, various exploratory tools can be used, along with methods that help identify causal relationships among the variables involved.

The development of a methodology for extracting causal relationships from social media discourse and real-world applications is crucial for advancing our understanding of how public sentiment and behavior evolve in response to key events and issues. By applying causal inference techniques to online discourse, we can identify how specific events, topics, or figures influence public sentiment, levels of engagement, and the spread of particular viewpoints. This approach not only allows for a better understanding of the dynamics of online conversations but also provides valuable insights for policymakers, organizations, and researchers who seek to anticipate and respond to public reactions. Moreover, the ability to establish causal relationships between public sentiment, specific topics and real-world events has the potential to inform strategies for addressing pressing global challenges, by identifying the key drivers that shape public opinion and discourse. The proposed methodology also contributes to the broader field of causal inference, offering a framework for analyzing complex social phenomena.

Association and causal rule mining on social media data has typically been employed to understand the underlying factors influencing certain phenomena, behaviors, or events. In particular, this kind of analysis is applied to understand the factors that lead to shifts in public sentiment over time, examining how external events or specific content can influence the mood or opinions of social media users (Trivedi, Patel & Acharya, 2018; Chandio & Sah, 2020; Fonseca et al., 2023).

The examination of social media discussions on issues related to climate change has gained attention from researchers across various fields, including data science (Hu, Huang & Wang, 2023; Schäfer & Hase, 2023), communication studies (Anderson, 2017; Mavrodieva et al., 2019), and sociology (Anderson, 2017; Pearce et al., 2019). However, a limited number of studies have looked into the issue of uncovering causal rules from climate change data.

Regression and correlation analysis have been commonly employed in the examination of social media data, primarily to investigate the relationship between two variables. However, despite their utility in predictive modeling, they leave the determination of clear causality open to question. In this type of analysis, relying solely on regression and correlation provides a preliminary understanding of the relationships between variables associated with climate change. The approach we propose seeks to deepen this understanding by examining temporal patterns, aiming to uncover not only the existence of associations but also the underlying mechanisms that drive them. The proposed approach focuses on extracting causal connections among variables representing topics, sentiment, aggressiveness, stance, and natural disaster events. Some additional discussion presented in this work aims to examine the impact of authority figures on public opinion in online settings. The main objectives of this research work are the following: O1 To propose and implement a new methodological pipeline that integrates data mining and causal inference techniques to detect patterns and causal relationships in online discourse and news-derived real-world events.

O2 To investigate the impact of specific topics, public figures, and real-world events on public sentiment and the spread of particular viewpoints.

O3 To contribute to the broader field of causal inference by offering a framework for analyzing complex social phenomena.

O4 To apply the proposed methodological pipeline to the specific case of climate change opinions and data.

The main contribution of this research is to provide a methodology for a comprehensive analysis of potential causal links between different variables in opinions and real-world events, going beyond simple correlations. The analysis also sheds light on the dynamics of online climate change discourse. The study uses publicly available datasets. The generated time series data and the code are provided as Supplemental Material to facilitate reproducibility.

The article is organized as follows. After introducing the research problem and its significance in the first section, the next section presents a review of related work. The methodology section outlines the proposed approach and presents the causal discovery algorithms employed. The methodological pipeline is then illustrated with an application to climate change. Finally, the conclusion section summarizes the key outcomes and proposes future research directions.

Related work

Many studies focus on sentiment and topic analysis to understand the relationship between social media discourse and real-world events (Sakaki, Okazaki & Matsuo, 2010; Bollen, Mao & Zeng, 2011; Preethi, Uma & Kumar, 2015; Pathak, Pandey & Rautaray, 2021; Uthirapathy & Sandanam, 2023). Most existing studies on sentimental causality rely on classical techniques, such as the Granger test and causal association rule mining. Several studies investigate sentiment analysis methods to understand public attitudes and opinions on specific topics through the analysis of social media content. For instance, Dahal, Kumar & Li (2019) apply topic modeling and sentiment analysis to a dataset of geotagged tweets related to climate change. The study demonstrates the use of volume analysis, topic modeling, and sentiment analysis (using VADER) to uncover public discourse patterns. Similarly, El Barachi et al. (2021) introduce a real-time sentiment analysis framework that employs a Bi-directional LSTM classifier to monitor public opinion on climate change.

In a comparative study, Sham & Mohamed (2022) evaluate various sentiment analysis approaches for classifying climate change tweets. The authors test seven lexicon-based approaches, three machine learning classifiers, and hybrid methods combining both. Another comparative study related to the analysis of Twitter climate action data (Rosenberg et al., 2023) explores diverse sentiment analysis methods, including lexicon-based techniques (VADER and TextBlob) and a bidirectional encoder representations from Transformers (BERT)-based transfer learning approach. The study highlights the superiority of BERT-based models over lexicon-based ones in terms of precision and accuracy. More recently, Anoop et al. (2024) emphasize the limitations of general language models in effectively representing climate-related text and stresses the importance of domain-specific models like ClimateBERT. Another recent work proposed by Diaz et al. (2024) presents the Polviz framework, which is designed to assess opinion polarization in social media, combining, topic modeling, visual analytics, generative artificial intelligence, and computational argumentation. Polviz is applied to address opinion polarization in climate change debates on Twitter.

The approaches mentioned above analyze the interactions between topics and emotions, offering valuable insights into sentiment analysis methodologies and their application to climate change discourse. However, they mainly focus on correlations and associations rather than explicitly addressing causal relationships. In contrast, this work aligns with recent literature that emphasizes moving beyond correlation, as proposed by Dong et al. (2021), by introducing a methodology aimed at extracting causal knowledge.

While causal analysis has been widely applied to social media data, most existing studies focus on areas other than climate change. In the financial domain, several studies have applied causal analysis to predict stock market behavior and product pricing trends. For instance, Granger causality has been used to demonstrate how sentiment polarity on social media can forecast stock price movements within days (Smailović et al., 2013) and how sentiment in online news articles affects oil prices with a lag of 3 weeks (Li et al., 2016). Transfer entropy has been employed to identify bidirectional causal links between social media sentiment and stock prices, highlighting the flow of information between these variables (Scaramozzino, Cerchiello & Aste, 2021). Causal analysis has also been applied to understand the interplay between social media sentiment, the financial sector, and political events. Research has shown that words used by politicians can affect stock markets (Silahtaroğlu, Dinçer & Yüksel, 2021), and public sentiment on issues like Brexit may influence currency exchange rates and market indices (Usher, Morales & Dondio, 2019). Examples of causal analysis applications in the political domain include the work of Fonseca et al. (2023), who propose a methodology for building causal graphs where nodes represent topics and emotions extracted from social media posts. Causal relationships have been explored in public health contexts, such as examining the impact of COVID-19 statistics on social media attitudes toward tourism (Godovykh et al., 2021). Additionally, causal analysis has been extensively applied to understand mental health issues by analyzing social media data (Yuan et al., 2023; Ogburn et al., 2024).

A study closely related to our work on extracting causal relationships from climate change social media data is proposed by Lenti et al. (2024). Their approach analyzes Reddit data from 2016 to 2022 using stochastic variational inference and Bayesian networks to examine the impact of sociodemographics, climate news exposure, extreme weather events, and social influence on activism. The findings reveal that media coverage and direct interactions with activists are key drivers of participation, particularly among left-leaning individuals and those from lower socioeconomic backgrounds. Our approach takes a broader perspective by integrating multiple causal inference techniques to examine causal relationships within a multi-source context. Furthermore, different from Lenti et al. (2024), we focus on public sentiment and discourse evolution over time, as well as real-world events, such as natural disasters, to explore how these dynamics interact with key topics related to climate change.

Another related study in the extraction of causal relationships from climate change discourse on digital media is the work presented by Chen et al. (2023). The authors analyze five million English tweets posted between 2018 and 2021 to explore how peaks in Twitter activity correlate with key events and how climate strike discourse has evolved. Additionally, they collected over 30,000 news articles from major English-speaking countries to examine the framing differences between media outlets and climate movement actors. Our work introduces a more comprehensive methodology for detecting relationships within climate change discourse. While the study by Chen et al. (2023) examines the patterns of discourse and framing differences between media outlets and climate movement actors, our approach uses a causal inference methodology that integrates multiple techniques to uncover links between public sentiment, discourse, and real-world events.

Methodology

This section presents a methodological pipeline composed of nine stages, designed to extract association rules and causal relations from opinions and real-world events, as shown in Fig. 1. The pipeline is designed to systematically analyze social media data and real-world event information to uncover meaningful patterns and relationships. It begins with the collection of topic-based data from social media platforms (stage 1) and real-world events from digital media sources (stage 2). This is followed by sentiment analysis to assess emotional tones, stance prediction and aggressiveness analysis to understand user positions and identify hostile language, and topic modeling to categorize discussions by subtopic within the main topic (stage 3). An exploratory analysis is conducted to identify initial patterns and insights (stage 4), as well as to determine appropriate thresholds for transforming continuous variables into categorical ones. The data is then preprocessed through cleaning, normalization, and variable selection to ensure consistency and quality (stage 5). The pipeline then employs frequent itemset mining and association rule mining to uncover relationships between variables (stages 6 and 7). Time series construction organizes the data temporally on a daily and weekly basis to analyze trends (stage 8). Finally, causal structure learning is applied to infer cause-and-effect relationships (stage 9).

Figure 1 Methodological pipeline comprising nine stages for extracting association rules and causal relations from opinions and real-world events.

Causality analysis

Causal analysis is a statistical approach used to identify and understand cause-and-effect relationships between variables. Unlike correlation and association rules, which only indicate a pattern of co-occurrence, causal analysis aims to reveal whether changes in one variable (the cause) directly influence changes in another variable (the effect). This method is particularly valuable in studies where understanding the directional impact of variables over time is critical, such as in environmental and social sciences. Causal analysis offers a richer perspective than exploratory methods that focus on co-occurring elements such as sentiments, topics, and stance. While these methods can identify patterns, they do not explain why these patterns occur or how variables interact. Causal analysis, on the other hand, uncovers directional relationships, allowing us to determine, for example, whether shifts in sentiment or stance are driven by specific events or if certain topics amplify aggressiveness in discourse.

When a ground truth is available, the validation of an inferred causal graph can be performed through multiple approaches. For instance, a structural comparison can be conducted using metrics such as the structural hamming distance (Acid & de Campos, 2003), which quantifies the number of edge additions, deletions, or reversals required to transform the inferred graph into the ground truth. Another metric, the structural intervention distance (Peters & Bühlmann, 2015), can be used to assess how well the inferred causal relationships align with the true causal structure by quantifying how interventions on one model affect the outcomes in the other model. Additionally, accuracy, precision, recall, and F1-score can be used to validate the inferred causal edges as done by Maisonnave et al. (2022). In those cases when ground truth is not available, certain graphs can be discarded if they include unrealistic or implausible causal links, based on domain knowledge or empirical evidence, as done by Maisonnave et al. (2024).

Five causal discovery algorithms are proposed for application to the generated time series. Four of them are selected based on the analysis reported in Maisonnave et al. (2022), which applies nine causal structure learning techniques to synthetic datasets and compares their performance. The algorithms that showed the best overall performance on synthetic data included Direct-linear non-Gaussian acyclic model (LiNGAM) (Shimizu et al., 2011), PC (Spirtes & Glymour, 1991), PCMCI (Runge et al., 2019), and vector autoregressive (VAR) (Sims, 1980), which are briefly described next. The remaining method, stochastic causality (Vinod, 2019), is included in our analysis because, in the application case described next, it produces sufficiently different results that, given the lack of ground truth, help prevent a potential “groupthink” interpretation.

Stochastic causality

Vinod (2019) introduces kernel causality by extending Granger’s ideas to situations where data is not necessarily a time series. He identifies the following limitation of Granger causality:“However, this is needlessly restrictive and inapplicable for human agents (who read newspapers) acting strategically at time t in anticipation of events at time t+1”.

To avoid assumptions of linearity, parametric definitions or temporal precedence in the causality analysis, Vinod constructs a test that exploits the asymmetries between Xi∼f(Xj) and Xj∼f(Xi), where f is a density function and ∼ indicates equivalence between the variables. This test relies solely on passively observed data, offering a more flexible and realistic method for causality analysis.

Vinod extends the concepts defined by Suppes (1970), where in contrast to the theory of deterministic causality, causality can be defined in a probabilistic manner, which tolerates noise and violations of the causal path between cause and effect. Formally, Xi causes Xj if P(Xj∣Xi)>P(Xj)a.e., where “a.e.” denotes almost everywhere. This can also be expressed in terms of conditional densities as f(Xj∣Xi)>f(Xj)a.e. The key idea in Vinod’s “Stochastic Kernel Causality” test is to exploit the asymmetries between f(Xj∣Xi) and f(Xi∣Xj) through kernel regressions. The test states that Xi causes Xj if the absolute errors in predicting Xj using Xi are smaller than the errors in predicting Xi using Xj, with control variables Xk.

Vinod proposes three criteria for constructing an index that measures the strength of the causal relations based on the residuals from the asymmetry analysis: (1) the model predicting Xk from Xi should outperform the model predicting Xi from Xk, (2) the absolute residuals of the first model should be smaller, and (3) the prediction accuracy (R-squared) of the first model should be higher.

Direct-LiNGAM

The linear non-Gaussian acyclic model (LiNGAM) (Shimizu et al., 2006) is a method used to identify causal relationships in datasets by assuming linear interactions among variables and a non-Gaussian distribution for exogenous (external) influences. In contrast to traditional linear models, LiNGAM leverages non-Gaussianity to distinguish causal directions by examining the statistical independence of residuals in regression analyses. Specifically, for a causal relationship X→Y, the linear regression Y=Xβ+ε should produce residuals ε that are independent of X. Conversely, if Y→X is the causal direction, the residuals from the regression of Y on X would show dependency, indicating a reversed causality.

LiNGAM-based approaches require that the causal relationships be linear and that the exogenous variables follow a non-Gaussian distribution, conditions that enable the model to resolve causal asymmetry from observational data alone. The Direct-LiNGAM (Shimizu et al., 2011) method improves upon earlier LiNGAM implementations by directly estimating causal relationships without first decomposing data into independent components, as in standard independent component analysis (ICA). Direct-LiNGAM iteratively estimates the causal ordering of variables and, under its assumptions, converges to the correct causal graph after a finite number of steps, equal to the number of variables. Additionally, it is robust to latent confounders, meaning it can manage unobserved variables that may affect both causal and outcome variables, enhancing its applicability in complex causal discovery tasks.

PC/PCMCI

The PC (Spirtes & Glymour, 1991) and PCMCI (Runge et al., 2019) methods are based on conditional independence testing. Both approaches employ partial correlation tests, estimating these correlations by applying Pearson’s correlation on residuals obtained from a linear regression between the variables of interest. For implementation, we use the ParCor function from the TIGRAMITE package (Runge et al., 2019).

The PC algorithm (Spirtes & Glymour, 1991) starts with a fully connected, undirected graph and progressively removes edges by testing for conditional independencies. Starting with 0th-order tests (no conditioning) and moving to higher orders (conditioning on one variable, then two, etc.), it reduces edges until it yields the undirected graph skeleton. In the second stage, the PC algorithm assigns directions to certain edges by identifying collider structures. This study employs a time series adaptation of the PC algorithm (Runge et al., 2019) which omits contemporaneous relationships.

PCMCI (Runge et al., 2019) operates in two stages as well: (i) it first applies a time series adaptation of PC to identify potential parent nodes for each variable at time t, denoted by Xtj, and (ii) it then uses the momentary conditional independence (MCI) test to examine whether Xt-τi→Xtj, where Xt-τi is a time-lagged parent of Xtj. The first stage thus constructs the Markov set for each node by removing irrelevant variables, while the second stage eliminates false positives from the identified relationships.

VAR

According to the notion of Granger causality (Granger, 1969), causality has two key properties: (1) the cause must precede the effect in time, and (2) the cause must contain unique information about the effect that is not shared by other variables. This concept of causality is based on autoregressive models, which are specifically designed for time series data, with the goal of detecting how the past values of a variable X may provide exclusive information about the future value of another variable Y.

The Vector Autoregressive (VAR) model (Sims, 1980) captures the relationships among multiple variables over time. In the VAR(p) variant, each endogenous variable is regressed against all other endogenous variables lagged from 1 to p periods. This represents each variable in terms of the past of all the others. The significance of the regression coefficients is then assumed to indicate the existence of a causal relation from the corresponding endogenous lagged variable to the response variable at the current period.

Application to climate change data

In this section, we apply the proposed methodological pipeline to investigate how climate change discussions on Twitter1 interact with sentiment, stance, and user behavior, as well as their potential links to external events such as natural disasters and political influences. To explore these interactions, we focus on three questions: Q1 What are the causal relationships between different climate change topics discussed on Twitter and the expressed sentiment (positive or negative), the stance (neutral, believer, or denier), and the aggressiveness (aggressive or not aggressive) of users engaging in those discussions?

Q2 Do specific natural disasters (e.g., storms, floods, extreme temperatures, wildfires, droughts) influence the sentiment, stance and the sometimes aggressive response of Twitter users towards the concept of climate change?

Q3 Is there a causal relationship between peaks in climate change denialism in public opinion and the presence of public figures known for being climate change denialists (De Pryck & Gemenne, 2017; Degani & Onysko, 2020; Byrne, 2020) in positions of power?

Pipeline stages and implementation

To answer the questions we apply the proposed pipeline to a combined dataset that includes variables extracted from tweets related to climate change, as well as other variables extracted from a database on natural disasters. Our analysis focuses on causality, but for completeness, we include frequent itemset and association rule analysis in Article S1.

Stage 1: topic-based data collection from social media

To illustrate the application of the proposed methodology we make use of the Twitter Climate Change Dataset (abbreviated as TCCD from now on) (Effrosynidis et al., 2022) (https://data.mendeley.com/datasets/mw8yd7z9wc/2). TCCD is constructed by combining three publicly available datasets: the Credibility of Climate Change Denial in Social Media Data (Samantray & Pin, 2019) (https://doi.org/10.7910/DVN/LNNPVD) (June 6, 2006–April 12, 2018), Climate Change Tweets IDs Data (Littman & Wrubel, 2019) (https://doi.org/10.7910/DVN/5QCCUU) (September 21, 2017–January 6, 2019 and April 18, 2019–May 17, 2019), and a filtered subset of the Twitter Archive Data (https://archive.org/details/twitterstream) (January 1, 2019–October 1, 2019). The analysis was completed only on tweets containing the hashtags #climatechange, #climatechangeisreal, #actonclimate, #globalwarming, #climatechangehoax, #climatedeniers, #climatechangeisfalse, #globalwarminghoax, or #climatechangenotreal. The final dataset used in this work consists of 15,789,400 unique tweet IDs spanning the period from January 1, 2007 to October 1, 2019. Although the original dataset of 15,789,411 tweet IDs begins on June 6, 2006, the sample size for the year 2006 (only 11 tweets) is too small to support meaningful conclusions. Consequently, that year was excluded from the analysis to avoid distorting the results.

Stage 2: real-world events collection

The International Disaster Database (EM-DAT) (Delforge et al., 2023) was used to enrich the dataset. EM-DAT is a comprehensive and authoritative source of global data on natural and technological disasters. It provides systematic, objective, and freely accessible information on the occurrence and impact of various types of disasters worldwide, including but not limited to earthquakes, floods, hurricanes, droughts, and technological accidents. The database includes information on the number of people affected, injured, or killed, as well as the economic damages caused by these events. Only natural disasters within the temporal scope of our dataset were considered, resulting in a total of 4,913 events.

Stage 3: data enrichment

Each tweet ID was enriched with four automatically generated labels, namely stance (neutral, believer, or denier), sentiment (values within the range −1 to 1, where negative values indicate negative sentiments and positive values correspond to positive sentiments), aggressiveness (aggressive or not aggressive), and topics (‘Weather Extremes’, ‘Importance of Human Intervention’, ‘Seriousness of Gas Emissions’, ‘Ideological Positions on Global Warming’, ‘Impact of Resource Overconsumption’, ‘Global stance’, ‘Politics’, ‘Significance of Pollution Awareness Events’, ‘Denialist Politicians vs. Science’ and ‘Undefined/One Word Hashtags’). Stance labels were predicted by a supervised BERT-based model (Devlin et al., 2018) trained on a dataset containing 34,667 manually labeled tweets. The sentiment value was determined through the use of two unsupervised lexicon-based models—VADER (Hutto & Gilbert, 2014) and Textblob (Loria, 2018)—as well as a pre-trained RNN model and a pre-trained BERT model implemented within the Flair framework (Akbik et al., 2019). Text aggressiveness information was predicted by a classifier trained on datasets provided by two competitions, namely Analytics Vidhya’s Twitter Sentiment Analysis competition (https://datahack.analyticsvidhya.com/contest/practice-problem-twitter-sentiment-analysis/) and Semeval 2021 task 7 (Meaney et al., 2021). Topics were obtained by applying Latent Dirichlet Allocation (LDA) (Blei, Ng & Jordan, 2003). Details of the method applied to generate the labels can be found in Effrosynidis et al. (2022).

Stage 4: exploratory analysis

Before processing the data we completed an exploratory analysis which shed light on several aspects of the dataset, such as the number of tweets and the values of the variables of interest over the years. The chart presented in Fig. 2 shows the tweet volume of the analyzed data set, represented by gray bars, and the years along the X-axis. Additionally, it displays the average values of three variables over time: average sentiment (in blue), average number of aggressive tweets (in red), and average number of denialist tweets (in green). Note that sentiment values have been mapped directly from the range [−1, 1] to [0%, 100%]. It is important to mention that the distribution of tweets about climate change across the years may not necessarily reflect reality but could instead be an artifact of the dataset used in this analysis. A significant portion of the tweets originates from the Climate Change Tweets IDs Data dataset, which provides an extensive list of tweets but is constrained to the September 21, 2017–January 6, 2019 and April 18, 2019–May 17, 2019 time frame. Although this limitation may influence the observed patterns in tweet volume over time, the chart enables us to see that average sentiment has generally remained neutral, while levels of aggressiveness and the percentage of denialist tweets have declined in recent years.

Figure 2 Tweets grouped by year, along with average sentiment values (mapped from [−1, 1] to [0%, 100%]), average number of aggressive tweets, and average number of denialist tweets.

The chart presented in Fig. 3 displays the distribution of tweets grouped by topic rather than by year. It can be observed that the topic ‘Global Stance’ has the highest number of associated tweets, exceeding 4 million, followed by ‘Importance of Human Intervention’ and ‘Weather Extremes’, each with approximately 2.5 million tweets. The topic with the highest percentage of aggression is ‘Politics’, exceeding 40%, which is not surprising, as individuals often exhibit increased aggression when discussing political issues. This is followed by the topic ‘Denialist Politicians vs. Science’, which unsurprisingly also displays the highest percentage of denialist tweets. Furthermore, the topic ‘Undefined/One Word Hashtags’ exhibits the most positive average sentiment, although it is challenging to analyze due to its lack of a clearly defined meaning. This is followed by the topics ‘Global Stance’ and ‘Importance of Human Intervention’.

Figure 3 Tweet volume grouped by topic, along with average sentiment values (mapped from [−1, 1] to [0%, 100%]), average number of aggressive tweets, and average number of denialist tweets.

For the natural disasters dataset, a simple chart was also generated to illustrate the distribution of various types of natural disasters over the years, as shown in Fig. 4. It is important to note that for this chart, the start date of the disaster was considered for simplicity, even though a disaster may have spanned multiple years. This graph highlights that there is a substantial number of disasters each year, exceeding 300 natural disasters annually. Furthermore, it is evident that in all years, there is a significant occurrence of storms and floods, both of which are types of disasters commonly associated with climate change.

Figure 4 Number of disasters grouped by year, with a color-coded classification indicating the volume of each disaster type.

Stage 5: data processing

Following the exploratory analysis, a data processing stage was undertaken. Initially, we filtered out data considered irrelevant or potentially introducing unnecessary noise. Subsequently, the data were discretized and grouped to facilitate the application of the required methods. Although TCCD, in its original form, includes data on latitude, longitude, temperature deviation, and gender, these variables were excluded from our analysis due to a significant proportion of missing values. Tweets associated with the topic ‘Undefined/One Word Hashtags’ were also excluded. Additionally, natural disasters that were considered not directly related to climate change were discarded. The disasters retained were those that could be associated with climate change: storms, floods, extreme temperatures, wildfires, and droughts. As a necessary step for the following stages, we performed discretization and grouping, specifically discretizing the sentiment variable, which was the only one with continuous values among the variables of interest. Values within the range [−1, −0.35] were mapped to ‘Negative Sentiment,’ values within the range (−0.35, 0.35) were assigned to ‘Neutral Sentiment,’ and values within the range [0.35, 1] were categorized as ‘Positive Sentiment.’ While other thresholds could be applied, these were chosen to ensure a balanced representation of sentiment categories and align with prior literature on sentiment classification.

Stages 6 and 7: frequent itemset mining and association rule mining

For completeness, we applied frequent itemset mining and association rule mining to explore co-occurrence patterns among variables in the dataset. While these techniques can reveal interesting associations within the discourse, they are not central to the current analysis, which focuses primarily on causal relations. Therefore, the results and methodological details of these stages are reported in the Supplemental Material (Article S1).

Stage 8: time series construction

Tweets were grouped by day to create a time series that clearly represents people’s opinions on each specific day. For simplicity, it was decided that the values of this time series should be binary (0 or 1), and the following variables were defined: Positive Sentiment: This variable takes the value 1 on a given day if the percentage of tweets labeled as ‘Positive Sentiment’ for that day exceeds a threshold (40%), and 0 otherwise.

Negative Sentiment: This variable takes the value 1 on a given day if the percentage of tweets labeled as ‘Negative Sentiment’ for that day exceeds a threshold (40%), and 0 otherwise.

Deniers: This variable takes the value 1 on a given day if the percentage of tweets labeled as ‘Denier’ for that day exceeds a threshold (20%), and 0 otherwise.

Aggressiveness: This variable takes the value 1 on a given day if the percentage of tweets labeled as ‘Aggressive’ for that day exceeds a threshold (40%), and 0 otherwise.

For each topic, a variable was created, which takes the value 1 on a given day if the percentage of tweets assigned to the corresponding topic for that day exceeds a threshold (25%), and 0 otherwise.

The threshold values chosen for each variable were established based on the average values of each variable. If a day has a value of 1 for the Positive Sentiment variable, it represents the fact that the percentage of tweets with positive sentiment on that day exceeds the average percentage of positive sentiment tweets across the entire dataset. Alternative approaches, such as minimizing entropy, tracking bursts, analyzing second derivatives, or using specific thresholds relevant to the field, could have been used for discretization to capture finer variations in sentiment, stance, and aggressiveness trends. However, we opted for a simplistic approach since, as observed in Fig. 2, the values remain mostly constant over time. Given this stability, a more intricate discretization method was not pursued, as it would not substantially alter the interpretation of the trends and could introduce unnecessary complexity in the analysis without a clear benefit.

Next, a time series was generated to characterize natural disasters. Once again, binary values were chosen for simplicity. Then, for each day in the time series, the variable associated with each type of natural disaster takes the value 1 if a disaster of that type was occurring on that day, and 0 otherwise. Finally, the time series were combined into a single time series that characterizes, for each day, both the way people express their opinions on Twitter and the ongoing natural disasters. The labels for the variables corresponding to topics were adapted from TCCD and prefixed with ‘[T]’ for clarity and ease of distinction (e.g., ‘[T] Weather Extremes’). Similarly, the labels for the variables representing natural disasters were prefixed with ‘[D]’ for clear identification (e.g., ‘[D] Flood’). Other similar time series were also created following the same process, but grouping tweets on a weekly or monthly basis instead of daily.

Stage 9: causal structure learning

In our analysis, we employ the five causal discovery techniques reviewed earlier to identify causal relationships among various tags associated with tweets and different natural disasters. This approach aims to detect patterns where changes in one variable lead, over time, to changes in another, providing deeper insights into the drivers of online climate discussions.

Insights from the pipeline application

The application of the proposed pipeline revealed distinct patterns and insights from the data, particularly in the causal analysis stage. The results of the five causal analysis methods vary considerably, with stochastic causality diverging notably from the other four (Direct LiNGAM, PC, PCMCI and VAR). To synthesize results from the latter, we constructed a unanimity ensemble, where a directed edge between two variables appears only if all four methods detect the same causal relation. Figures 5 and 6 show the resulting causal graphs for daily and weekly frequencies, respectively. Figure 7 shows the result of stochastic causality for daily data. Figures 8, 9, 10 and 11, depict the results of splitting the datasets in two periods, 2007 to 2014 and from 2015 to 2019, to highlight the appearance of public figures widely seen as supporting some version of climate denialism (see De Pryck & Gemenne (2017), Degani & Onysko (2020) or Byrne (2020)) on the political stage. Notice that all these methods are sensitive to the change in frequency of the data. This affects the answers to our questions. We might speculate that long-term relationships may be more relevant to answer Q1, Q2, and Q3, since these inquiries involve reactions that may extend over time.

Figure 5 Unanimity ensemble model for daily data (PC-PCMCI-DirectLingam-VAR).

Figure 6 Unanimity ensemble model for weekly data (PC-PCMCI-DirectLingam-VAR).

Figure 7 Stochastic causality results for daily data, 2007–2019.

Figure 8 Stochastic causality results for daily data, 2007–2014.

Figure 9 Stochastic causality results for weekly data, 2007–2014.

Figure 10 Stochastic causality results for daily data, 2015–2019.

Figure 11 Stochastic causality results for weekly data, 2015–2019.

Although no relationships are supported by both the unanimity ensemble and stochastic causality across daily and weekly frequencies, two common links appear in their daily frequency graphs (Figs. 5 and 7, respectively): ‘[T] ImportanceofHumanIntervention’→‘[T] GlobalStance’ and‘Deniers’→‘NegativeSentiment’. A key distinction between stochastic causality and the unanimity ensemble of Direct LiNGAM, PC, PCMCI and VAR is that the latter includes untenable causal relations between topics or sentiments in Twitter and some natural disasters. Although there is no established ground truth for the causal relationships in our dataset, certain causal directions can be deemed implausible, such as Twitter sentiments causing natural disasters. Stochastic causality’s ability to avoid such erroneous associations reinforces its credibility and highlights its value in supporting more reliable answers to our research questions.

To address question Q1, which explores the causal relationships between different climate change topics discussed on Twitter and the expressed sentiment, stance, and aggressiveness of users, we find that the causal relationships derived from stochastic causality generally show a flow from topics to sentiment, stance, and aggressiveness (Figs. 7–11). This indicates that topics tend to influence discourse tone rather than the other way around. We observe that topics that evoke strong emotions, particularly negative ones, could lead to increased aggressiveness and reinforce existing sentiments or stances, as is the case of ‘[T] Weather Extremes’, ‘[T] Denialist Politicians vs. Science’, and ‘[T] Seriousness of Gas Emissions’. This observation aligns with studies that highlight the causal impact of emotionally charged topics on the tone and polarization of online climate discourse (Bassolas et al., 2024).

Our second question Q2 examines whether there exists a strong causal relationship between the occurrence of disasters and the discussions in the Tweetosphere. The evidence provided by stochastic causality indicates that global discussions on Twitter remain disconnected from specific localized real-world climate events. It is worth mentioning that this finding is consistent with previous work suggesting that political events and media coverage, rather than actual climate events, play a more substantial role in shaping public discourse on social media platforms (Schäfer, 2012; Wang, Song & Chen, 2023).

The independence of global discussions from localized world events hints at a potential negative answer to our third question Q3, about the relationship between denialism and the accession of denialist politicians to office. Focusing again solely on stochastic causality, we examine the relationship between variables ‘[T] Denialist Politicians vs. Science’ and ‘Deniers’. While a significant correlation between these two variables exists both before and after the candidacy of an influential denialist figure for the 2016 US presidential election, it is not significant over the entire period under analysis. This can be explained by the change in the Pearson correlation coefficient between the variables in the 2007–2014 and 2015–2019 periods: 0.28 and 0.43, respectively. In turn, for the entire period, it drops to just 0.08. Interestingly, the impact of the topic ‘[T] Denialist Politicians vs Science’ shifts slightly between the periods 2007–2014 and 2015–2019. Stochastic causality detects a causal influence on ‘Positive Sentiment’ at the daily frequency and on ‘Negative Sentiment’ at the weekly frequency during 2007–2014 (Figs. 8 and 9). However, in 2015–2019, the topic ‘[T] Denialist Politicians vs Science’ simultaneously has a causal influence on both Positive and Negative Sentiment (Figs. 10 and 11). The observed weak and inconsistent causal links between the political rise of denialist figures and stance or sentiment contribute to the long-standing discussion on the indirect and mediated relationship between political outcomes and public opinion in the climate debate (Dunlap & McCright, 2011; Gounaridis & Newell, 2024).

Conclusions and future research

In this work we proposed a methodology to analyze the relationship between interactions on social media and external events, using climate-change data as a case study. Specifically, we examined whether the information about such events influenced the discourse on the social network and examined the potential impact of a disruptive political figure with controversial views on climate change. Our analysis used five different causal detection methods, with particular emphasis on the insights provided by stochastic causality.

Our findings highlight the challenges of detecting robust causal relationships in social media data. While Twitter is often seen as the “pulse of the world,” reflecting and amplifying global conversations in real-time, stochastic causality did not reveal strong evidence of a global connection between the Tweetosphere and natural disasters. This does not rule out the possibility of such relationships emerging in localized contexts, where Twitter activity might more closely correspond to specific events. Future research could refine the methodology by focusing on these localized dynamics to better understand how regional interactions between social media and real-world events manifest.

On the other hand, the ascension of denialist political figures seems to have had little impact on the topics and sentiments examined here, having only impact on both negative and positive sentiments and on aggressiveness. While these results do not seem to shed much light on how the public discussion on climate change have evolved during the last two decades, they allow us to infer that, perhaps, Twitter exhibits a certain disconnection of some of the important issues affecting the planet, being more concerned with its endogenous motivations. As part of future research, a more granular analysis could be conducted to investigate local or regional contexts where Twitter activity may align more closely with specific climate-related events or discussions. Exploring the role of influential accounts, hashtags, and coordinated campaigns could provide deeper insights into how climate-related topics capture public interest.

While integrating multiple datasets enriches the analysis by providing a more comprehensive view, it may also introduce data loss and alter sample composition. Although we conducted an exploratory data analysis and applied cleaning and normalization techniques to ensure consistency, a more sophisticated analysis could be performed by comparing key distributions before and after integration. Since some degree of sampling bias may still exist, future research could systematically evaluate these distributional changes to better quantify and mitigate potential biases.

One of the limitations of our work is related to the recent changes in the accessibility of X (formerly Twitter) data, which have affected the ability to collect and analyze large-scale social media content. Additionally, the reduction or truncation of specific time periods, can potentially limit the ability to generalize findings and make comparisons across different time spans. To partially overcome this limitation, future research could explore alternative platforms such as Reddit, Bluesky, Mastodon, and other decentralized networks, which still offer opportunities for large-scale data collection. Additionally, e-participation platforms, which allow citizens to engage in political and social discourse online, could serve as valuable sources of data for analyzing public sentiment and its evolution in response to key events.

On the general question of the usefulness of causal detection methods, we can see that the lack of a ground truth makes it difficult to assess the validity of the results of those methods. But, at least for the data analyzed in this article, stochastic causality seems to yield results marginally more cogent. More work is needed to see if this claim can be generalized to other datasets and whether in other fields a consensus of all the methods could be obtained. Future research could focus on establishing benchmark datasets with well-defined causal ground truths to better evaluate the effectiveness of various causal detection methods. However, this task is not straightforward, as ground truth may not always be readily available, particularly in complex systems where causal relationships are difficult to observe or measure directly. Lastly, developing techniques to integrate domain expertise with causal inference methods could enhance the interpretability and relevance of the findings.

Supplemental Information

Supplemental Information 1 Climate Change Time Series Data.

Normalized climate change time series data with raw data files including information from tweets and natural disasters at daily and weekly frequencies for the period 2007–2019.

Supplemental Information 2 Exploratory and Causal Analysis - Climate Change Jupyter Notebook.

The code for data processing, exploratory analysis, frequent itemset mining, association rule generation, and causal analyses using VAR, LiNGAM, PC, PCMCI, and ensemble method.

Supplemental Information 3 R code for stochastic causality on Climate Change time series.

R code for completing the stochastic causality analysis.

Supplemental Information 4 Frequent Itemset and Association Rule Mining.

An overview of frequent itemset and association rule mining. Application to climate change data analysis.

Additional Information and Declarations

Competing Interests

Ana G. Maguitman is an Academic Editor for PeerJ.

Author Contributions

Juan Marten conceived and designed the experiments, performed the experiments, analyzed the data, performed the computation work, prepared figures and/or tables, authored or reviewed drafts of the article, and approved the final draft.

Fernando Delbianco conceived and designed the experiments, performed the experiments, analyzed the data, performed the computation work, prepared figures and/or tables, authored or reviewed drafts of the article, and approved the final draft.

Fernando Tohme conceived and designed the experiments, performed the experiments, analyzed the data, prepared figures and/or tables, authored or reviewed drafts of the article, and approved the final draft.

Ana G. Maguitman conceived and designed the experiments, performed the experiments, analyzed the data, performed the computation work, prepared figures and/or tables, authored or reviewed drafts of the article, and approved the final draft.

Data Availability

The following information was supplied regarding data availability:

The raw data and code are available in the Supplemental Files.

The Climate Change Twitter Dataset is available at Mendeley: Effrosynidis, Dimitrios (2022), “The Climate Change Twitter Dataset”, Mendeley Data, V2, doi: 10.17632/mw8yd7z9wc.2.

The Credibility of Climate Change Denial in Social Media Data is available at Harvard Dataverse: Samantray, Abhishek; Pin, Paolo, 2019, "Data and code for: Credibility of climate change denial in social media", https://doi.org/10.7910/DVN/LNNPVD, Harvard Dataverse, V1.

The Climate Change Tweets IDs Data is available at Harvard Dataverse: Littman, Justin; Wrubel, Laura, 2019, "Climate Change Tweets Ids", https://doi.org/10.7910/DVN/5QCCUU, Harvard Dataverse, V1

A filtered subset of the Twitter Archive Data is available at: https://archive.org/details/twitterstream.

1 In the rest of this article we will keep using the former name of this social network since the dataset of tweets used in our analysis corresponds to a period previous to its change of name.

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
