# Peer review of "A methodological approach for inferring causal relationships from opinions and news-derived events with an application to climate change"

_PeerJ Computer Science, doi:10.7717/peerj-cs.2964_

## Round 0.1 · original submission · Major Revisions

Your paper presents valuable methodological contributions, but the reviewers highlight key concerns that need to be addressed. The primary issue is the unclear positioning between methodological and applied contributions—clarifying this and adjusting the framing accordingly is essential. Both reviewers question the binarization of sentiment and discourse variables, particularly the choice of threshold values, which require stronger justification or sensitivity analyses. Graphical clarity (e.g., Figure 1 and network visualizations) should be improved, and potential biases from data integration should be acknowledged. Additionally, comparing your approach with prior studies on climate discourse and causal inference would strengthen the paper. Lastly, discussing recent changes in X (Twitter) data accessibility and their implications would add valuable context.

·

Basic reporting

The article examines the relevant problem of identifying causal relationships between social reaction and such an important social agenda as climate change issues. The article is written in fairly good English. The text is adequately structured.
A general comment is the dual positioning of the article. On the one hand, the research questions formulated by the authors are of an applied essence and are related to determining the nature of the relationships between the public reaction of the social media audience to the problems of climate change, natural disasters and the opinions of authoritative figures. On the other hand, the main contribution of the article is declared to be the research methodology. That is, the article is ultimately positioned by the authors as a methodological one, but the research questions are of an applied nature. It is necessary to resolve this contradiction. It should be noted that the technology of data preprocessing implemented by the authors includes their binarization by means of threshold values. This leads to a coarsening of the initial data and the emergence of questions about the reliability of the obtained results in the applied sense. Therefore, positioning the work as a methodological one seems correct, but in this case, a corresponding adjustment of the research questions and, possibly, the general structure of the article is required.
Some specific comments:
- Fig. 1 contains data for 2006, which, according to the authors themselves (lines 225-227), has a limited sample size. And the curves have a striking sharp decline, which, at the same time, cannot be adequately interpreted due to the limited data. It would be better to limit the abscissa axis to a time range where the data volumes are comparable and the shape of the obtained curves can be interpreted.
- In figures 9, 10, 12, 13, the arcs on the graphs have excessive intersections and are poorly traced. For better perception of the figures, I recommend adjusting the location of the vertices and arcs to remove unnecessary intersections.

Experimental design

The experimental part of the work is described in sufficient detail. The sources and data cleaning and preprocessing steps are fully described.
However, serious questions arise regarding the data preprocessing algorithms used by the authors. For use in the considered algorithms of causal analysis, the data are binarized using threshold values. It seems that such an approach greatly affects the reliability of the results in the applied sense - the choice of other threshold values can give a completely different result. The authors provide a rationale for the choice of threshold values, but these values were chosen relative to the average indicators for the entire sample. The adequacy of this approach to determining threshold values requires argumentation, since the average value of public discourse parameters (for example, the middle level of aggression) can change significantly in different observation periods. In such conditions, tracking bursts, "second derivatives", parameter values corresponding to the expected consequences seems to be a more correct approach to identifying the cause-and-effect relationship.

Validity of the findings

As noted above, the reliability of the conclusions is questionable due to the technology used for pre-processing the initial data. In essence, the pre-processing algorithm used formalizes the parameters of public discourse in the form of some model. It seems that in order to substantiate the conclusions regarding the presence of cause-and-effect relationships between the parameters of discourse and external factors, it is first necessary to substantiate the adequacy of the discourse model used. An alternative solution to this issue may be to change the focus of the article towards methodological results, but in this case the object of study and comparative analysis should be the algorithms used and the technology of causal analysis as a whole.

Additional comments

The article presents the results of a large experimental work aimed at solving a relevant and non-trivial problem of significant practical importance. The results of the study are of considerable interest both from the point of view of practical use in the analysis of social processes based on social media data, and for the development of a methodologies for analyzing such processes using modern methods of information processing.
The main weak point of the work is the duality of goal setting in the direction of simultaneously obtaining applied and methodological results.

Reviewer 2 ·

Basic reporting

The paper is generally easy to follow and well-written. However, I believe it lacks detailed explanations for several research design choices, which could be crucial for understanding the study's methodology. A particular concern is the decision to convert sentiment analysis into a binary variable based on a 40% threshold. The rationale for selecting this specific cut-off point, or why a binary representation was deemed suitable for this variable, is not provided. This omission raises concerns about whether this methodological choice could potentially influence the study's results. For instance, the definition provided—"Positive Sentiment: This variable is assigned a value of 1 on a given day if the percentage of tweets labeled as ‘Positive Sentiment’ exceeds 40%, and 0 otherwise"—lacks justification for the threshold choice, which is vital for assessing the robustness of the findings. Additionally, to enable a fair comparison with previous literature, I recommend that the author collects and uses the exact measurements of features utilized in prior studies. Analyzing their potential causal relationships could provide deeper insights and enhance the validity of the comparisons.

While I may not be an expert in the topic area, from a generalist's perspective, I have a question about the inclusion of general causal inference techniques in the literature review. Specifically, if these techniques are broadly applicable, what makes their application in the context of climate change noteworthy? It seems the paper aims to make a methodological contribution, but I am not entirely convinced of this. One approach the author could consider is comparing their results with other findings related to climate change opinions on social media to identify any discrepancies. Additionally, it might be beneficial to review studies that do not employ causal inference techniques. This could position the paper as a kind of meta-review, helping the research community to identify and possibly reevaluate unsupported or anomalous findings in existing literature.

The author's approach to combining multiple datasets is great. However, such combinations can often lead to data loss and changes in the sample composition. It would be beneficial for the author to address these issues, ensuring that the data integration does not introduce sampling bias, or alternatively, to discuss this aspect in the limitations section. Additionally, I appreciate the author's transparency in sharing the complete, runnable code and data.

I also suggest the author includes a discussion on the recent updates related to access to X data and the ongoing trends in climate change and political discourse. How might these changes (the length of the data) impact the findings of the study?

Experimental design

please refer to the basic reporting section

Validity of the findings

Please refer to the Basic Reporting section.

Additional comments

Please refer to the Basic Reporting section.

---

## Round 0.2 · Minor Revisions

Remaining concerns are mainly about the framing of the paper and other minor concerns. As Reviewer 2 pointed out, the paper employs existing methods and thus is not exactly a "method paper". Still, the paper focuses on the methodological aspects given how these methods are combined together and tested. In addition to addressing all concerns, regarding framing of the paper, I'd recommend the authors to focus on the development & testing of a "pipeline" applicable to social media data using a case study of climate change as the primary contribution of the paper.

·

Basic reporting

The paper is written in fairly good English. The introduction provides a comprehensive presentation of the context of the study, and the text of the article is provided with links to recent articles on the relevant topic. The text of the manuscript is structured in accordance with two main objectives of the research – the development of a methodological approach to the causal analysis based on heterogenous data using several techniques, and its case study using the example of climate change. The material is accompanied by the original raw data used in the study. The main stages of the study and the results are illustrated with sufficiently informative figures.
However, some of them may be subject to comments. Thus, Figure 2, despite the authors' remarks within the text, provokes a not entirely correct interpretation of the data. The main visual metaphor of the line graph, in my opinion, is to illustrate at a qualitative level the dynamics of one value or the mutual ratio of several. In this sense, the figure suggests that in 2006-2007 there was a dramatic decrease in the aggressiveness of tweets. However, in the text of the article, the authors make a remark that due to the small number of observations in this period, the reliability of this conclusion is questionable. In Figure 10, the edges between the pairs of nodes “Politics”-“Positive Sentiment” and “Importance of Human Interventions”-“Negative Sentiment” are only discernible at high magnification. And in Figure 7, the edges originating from the nodes “Importance of Human Interventions” and “Global Stance” in the direction of the nodes “Positive Sentiment” and “Negative Sentiment” are not discernible at all.
These comments are not crucial, but taking them into account can improve the perception of the manuscript.

Experimental design

The work has a methodological nature, its main contribution is the development of a methodology for studying complex social phenomena using a set of methods for analyzing causal relationships. The objectives of the work and the research questions are formulated quite clearly. The methodology of the work is described in sufficient detail. The sources of the dataset used as well as data cleaning and preprocessing steps are fully described.

Validity of the findings

The main conclusions of the work are the statement of the complexity of the task of studying causal relationships based on social media and external data. The experiments conducted by the authors demonstrated the advantage of the Stochastic Causality in comparison with other methods of analysis on the used dataset. However, the authors rightly note that the lack of ground truth data does not allow for unambiguous conclusions. In general, the work provides a fairly in-depth analysis of the methodological problems of analyzing causal relationships on heterogeneous data using different methods. The conclusions drawn from the work are sufficiently substantiated, and the authors outline the directions for further research on this issue.

Additional comments

The strong point of the work is that the authors consider a fairly wide range of methods for studying causal relationships, including modern deep learning models, as applied to a challenging applied problem. The results obtained in the work can serve as a basis for further research on this topic.

Reviewer 2 ·

Basic reporting

I want to thank the author for the significant effort put into the revision; the quality of the paper has improved. However, my concerns still center around the framing of the work. The author has reframed the paper as a methodology paper, but I question this characterization. Both the data mining and causal inference methods used are well-established, and the author primarily applies these methods to a domain. In my view, this represents an applied contribution rather than a methodological innovation. I recommend that the author emphasize the applied aspect more clearly and highlight what new insights or results emerge from applying these methods. It would also strengthen the paper to provide examples where the results support or extend existing theories.

Experimental design

see above

Validity of the findings

see above

---

## Round 0.3 · accepted · Accept

The authors have addressed all comments and the manuscript is ready for publication.

·

Basic reporting

The article is written in clear and unambiguous, professional English. Considered research background and findings of the work are well referenced with relevant literature provided. The introduction provides a comprehensive presentation of the context of the study, and the text of the article is provided with links to recent articles on the relevant topic. The text of the manuscript is structured in accordance with two main objectives of the research – the development of a methodological approach to the causal analysis based on heterogenous data using several techniques, and its case study using the example of climate change. The material is accompanied by the original raw data used in the study. The main stages of the study and the results are illustrated with sufficiently informative figures. The figures are provided with the necessary comments in the text.

Experimental design

The work has a methodological nature; its main contribution is the development of a methodological pipeline for studying complex social phenomena using a set of methods for analyzing causal relationships. The objectives of the work and the research questions are formulated quite clearly. The methodology of the work is described in sufficient detail. The sources of the dataset used as well as data cleaning and preprocessing steps are fully described. The proposed methodology is tested on the issue of climate change, which is very relevant both in terms of environmental problems as such and in terms of the impact on public sentiment. The originality of the approach is demonstrated by the fact that the authors jointly analyze both natural facts related to natural disasters and the impact of public opinion influencers.

Validity of the findings

The main conclusions of the work are the statement of the complexity of the task of studying causal relationships based on social media and external data. The experiments conducted by the authors demonstrated the advantage of the Stochastic Causality in comparison with other methods of analysis on the used dataset. However, the authors rightly note that the lack of ground truth data does not allow for unambiguous conclusions. In general, the work provides a fairly in-depth analysis of the methodological problems of analyzing causal relationships on heterogeneous data using different methods. The conclusions drawn from the work are sufficiently substantiated, and the authors outline the directions for further research on this issue.

Additional comments

The strong point of the work is that the authors consider a fairly wide range of methods for studying causal relationships, including modern deep learning models, as applied to a challenging applied problem. The results obtained in the work can serve as a basis for further research on this topic, as well as a pipeline for cause-relationships analysis on the base of heterogeneous data in another subject areas.